# Late-Presenting Swimming-Induced Pulmonary Edema: A Case Report Series from the Norseman Xtreme Triathlon

**DOI:** 10.3390/sports7060137

**Published:** 2019-06-03

**Authors:** Jørgen Melau, Martin Bonnevie-Svendsen, Maria Mathiassen, Janne Mykland Hilde, Lars Oma, Jonny Hisdal

**Affiliations:** 1Institute of Clinical Medicine, University of Oslo, 0316 Oslo, Norway; jonny.hisdal@medisin.uio.no; 2Department of Vascular Surgery, Oslo University Hospital, 0424 Oslo, Norway; martin.bonnevie@gmail.com; 3Prehospital Division, Vestfold Hospital Trust, 3103 Toensberg, Norway; 4Department of Cardiology, Telemark Hospital Trust, 3710 Skien, Norway; maria.mathiassen@gmail.com; 5Department of Cardiology, Akershus University Hospital, 1478 Loerenskog, Norway; janne.mykland.hilde@ahus.no; 6Baerum Hospital, Vestre Viken Hospital Trust, 3004 Drammen, Norway; lars.oma@mac.com; 7Institute for Surgical Research, Oslo University Hospital, 0424 Oslo, Norway

**Keywords:** SIPE, endurance, triathlon, open water, swimming

## Abstract

Swimming-induced pulmonary edema (SIPE) may develop during strenuous physical exertion in water. This case series reports on three cases of suspected late-presenting SIPE during the Norseman Xtreme Triathlon. A 30-year-old male professional (PRO) triathlete, a 40-year-old female AGE GROUP triathlete and a 34-year-old male AGE GROUP triathlete presented with shortness of breath, chest tightness and coughing up pink sputum during the last part of the bike phase. All three athletes reported an improvement in breathing during the first major uphill of the bike phase and increasing symptoms during the downhill. The PRO athlete had a thoracic computed tomography, and the scan showed bilateral ground glass opacity in the peripheral lungs. The male AGE GROUP athlete had a normal chest x-ray. Both athletes were admitted for further observation and discharged from hospital the following day, with complete regression of symptoms. The female athlete recovered quickly following pre-hospital oxygen treatment. Non-cardiogenic pulmonary edema associated with endurance sports is rare but potentially very dangerous. Knowledge and awareness of possible risk factors and symptoms are essential, and the results presented in this report emphasize the importance of being aware of the possible delayed development of symptoms. To determine the presence of pulmonary edema elicited by strenuous exercise, equipment for measuring oxygen saturation should be available for the medical staff on site.

## 1. Introduction

Swimming-induced pulmonary edema (SIPE) is a potentially dangerous complication during strenuous exercise with the possibility of accompanying misdiagnosis. Cough, dyspnea, hemoptysis and hypoxemia may develop after surface swimming or diving, often in young, healthy individuals. The condition is relatively rare, but an increasing number of cases of SIPE is being reported in triathletes [1]. Episodes are more likely to occur in highly fit individuals undertaking strenuous or competitive swims. A study of American triathletes reported an association between SIPE and hypertension [1]. Other potential risk factors included diabetes, fish oil use, long course triathlons, wearing a wetsuit and female sex [1]. SIPE usually resolves spontaneously within 24 hours, or with beta2 adrenergic agonist or diuretic therapy, but can be fatal if not treated [2,3]. Individuals who develop SIPE often have recurrences under the same conditions [4,5]. Increased awareness among health professionals and organizers, in addition to early detection of symptoms of SIPE, are therefore warranted to avoid life-threatening episodes during future extreme triathlon competitions.

Previous case reports have described cases with sudden onset of shortness of breath during swimming, and shortly after the swimming phase of a triathlon resulting in withdrawal from the race [6,7,8,9]. There are also reports of athletes completing entire triathlon events with symptoms of SIPE during the bike and run legs [10]. In this case series, we report three cases of slowly progressing pulmonary edema with resultant race withdrawal, one after 122 km and two cases after 180 km, respectively, during the Norseman Extreme Triathlon in 2016, 2017 and 2018. All three cases were diagnosed with swimming-induced pulmonary edema (SIPE) and displayed an undulating symptom behaviour not previously described in the literature. Two athletes were admitted to hospital for further examination, and one athlete did not need hospitalization.

## 2. Case Report

A 30-year-old male PRO triathlete, a 40-year-old female AGE GROUP athlete and a 34-year-old AGE GROUP athlete, all with no history of medical illness, who regularly competed at long distance triathlons presented with shortness of breath, chest tightness and coughing up pink sputum during the last part of the bike phase of a full distance triathlon race. Ambient air temperature varied between 6 and 8 °C and the water temperature was around 14.2 to 14.4 °C during the races in 2016 and 2017 and somewhat higher in 2018 with an air temperature of 10 °C and water temperature at 17.5 °C. The PRO male athlete had completed the same competition the previous 3 years without any medical complications. The female and the male AGE GROUP athletes entered their first Extreme triathlon, but they had both previously completed several full Ironman competitions without symptoms of SIPE. As part of another study during the same competition, normal spirometry was performed the day before the race for both the PRO athlete and the female athlete. Written informed consent for publication was obtained from all athletes.

During the swim, all athletes wore a well-fitted wetsuit and neoprene swim cap. They later reported that they felt increasingly breathless and were struggling to keep the usual pace in the water just minutes after the start. None of them reported that they had aspirated during the swim. 

The bike leg during the Norseman Extreme Triathlon starts with approximately 40 km uphill, from sea level to 1245 m above sea level (Figure 1). 

All athletes reported an improvement in breathing during the first major uphill (1245 m elevation over 40 km) of the bike phase and reported increasing symptoms during the downhill. The female athlete contacted the medical crew due to shortness of breath and coughing pink sputum after about 122 km. During the clinical examination, she presented mild facial edema, jugular vein distention and lip cyanosis. Peripheral capillary oxygen saturation (SpO_2_) was measured to 88%, and she was coughing up pink sputum. No signs of airway obstruction were found, and she denied any previous history of allergies. The athlete was taken out of the competition by one of the race doctors. She was transferred to the nearest local medical center for further treatment but recovered quickly after being treated with oxygen and hospitalization was not indicated. The male athletes managed to complete the 180 km long bike phase, but due to coughing and increased shortness of breath, they were both forced to withdraw from the race just a few meters into the run phase. 

The PRO athlete was examined by one of the race doctors immediately after withdrawal. The medic reported end-expiratory crackles bilaterally and a productive cough with pink sputum. He had facial edema but no cyanosis. He was transferred to the nearest hospital, approximately 1 h away for further examination. 

On arrival, his blood pressure was 105/62 mmHg, respiratory rate 20 breaths/min, heart rate 84 beats/min and oxygen saturation was 96% on room air. A laboratory examination showed a serum sodium level of 142 mEq/L, B-natriuretic peptide (BNP) level of 26 pmol/L (normal range < 10 pmol/L), troponin T 76 ng/L (normal range < 10 ng/L), C-reactive protein 23 mg/L (normal range < 5 mg/L), hemoglobin 12.1 g/dL and D-dimer 0.7 mg/L (normal range < 0.5 mg/L). An electrocardiogram (ECG) showed sinus rhythm with incomplete right bundle branch block (RBBB). He had normal findings on clinical examination.

Arterial blood gas results were; pH 7.48, pCO_2_ 4.84 kPa, pO_2_ 9.39 kPa, bicarbonate 26.2 mmol/L, base excess 3.4 mmol/L and lactate 2.7 mmol/L. 

An echocardiogram was performed at arrival at the emergency department that showed a well-contracting, normal-sized left ventricle and mildly dilated right ventricle. The end-diastolic impression of the intraventricular septum led to suspected high pulmonary artery pressure.

At the hospital, he was initially diagnosed with suspected pulmonary embolism and transferred to a larger regional hospital for a thoracic computed tomography (CT) scan. The findings included bilateral ground glass opacity in the peripheral lungs but no pulmonary embolism. Due to the complete regression of symptoms, no treatment other than rehydration and oxygen was given during hospitalization. He was discharged from hospital the following day.

At a 1-month follow-up-examination, the athlete had fully recovered with complete regression of CT findings and echocardiogram changes. He was diagnosed with swimming-induced pulmonary edema.

The male AGE GROUP athlete called the emergency number directly and was not examined by a race doctor. After completing the bike phase (180 km), he experienced increased shortness of breath and coughing with pink sputum. When examined by the paramedics he presented with heavy coughing. SpO_2_ was measured to 90%. He was transferred to the nearest hospital for further examinations, which was the same hospital as the PRO athlete was transferred to two years earlier. On arrival, his blood pressure was 123/70 mmHg, respiratory rate 16 breaths/min, heart rate 70 beats/min and oxygen saturation was 93% on room air. A laboratory examination showed a serum sodium level of 142 mEq/L, troponin T 101 ng/L (normal range < 10 ng/L), C-reactive protein 8 mg/L (normal range < 5 mg/L), hemoglobin 13 g/dL and D-dimer 0.4 mg/L (normal range < 0.5 mg/L). An ECG showed sinus rhythm with pathological R-progression in V1 to V6. Clinical examination reported end-expiratory crackles bilaterally but otherwise normal findings.

Arterial blood gas results were; pH 7.46, pCO_2_ 5.01 kPa, pO_2_ 8.20 kPa, bicarbonate 25.9 mmol/L, base excess 2.6 mmol/L and lactate 2.2 mmol/L. 

No echocardiogram was performed. He was given diuretics and was discharged from the hospital with complete regression of symptoms the following day.

## 3. Discussion 

All three athletes presented in this report experienced a debut of shortness of breath during the swim phase of an extreme triathlon competition with symptomatic progression throughout the following bike phase. Upon presentation for medical personnel, all athletes displayed decreased levels of oxygen saturation, of which two were measured pre-hospital. Pulmonary edema would negatively impact ventilation and thereby, oxygen saturation. In the presented cases, the use of on-site pulse oxiometry served to strengthen the suspicion of potential SIPE.

During hospital examination, the PRO athlete had several blood biomarkers outside of the normal range. In our experience, results of not yet published data suggest this is a common and potentially normal finding in asymptomatic participants of extreme triathlons. His RBBB was considered a normal variant as often seen in young, well-trained athletes. Following an echocardiogram ruling out cardiac failure, and CT scan revealing no signs of emboli but the presence of ground glass opacity, the clinical picture best fits that of SIPE. 

The ECG of the AGE GROUP athlete displayed pathological R-progression in V1 to V6, which was also considered a normal variant in well-trained endurance athletes. Due to his elevated troponin levels, ischemic cardiac disease cannot be ruled out entirely as a potential etiology. However, this athlete had no known risk factors for ischemic heart disease, and this differential diagnosis would not account for symptomatic improvement during uphill cycling. As such, the overall clinical presentation is arguably more suggestive of SIPE. 

To the best of our knowledge, the female athlete did not undergo diagnostic blood tests or radiographic examinations. Therefore, the possibility of other differential diagnoses, such as angio-oedema or asthma attack, cannot be ruled out entirely. Nevertheless, we would argue that the absence of obstructive symptoms, no prior history of allergies, and quick symptom regression following oxygen treatment suggests SIPE is the more likely diagnosis.

Despite the early presentation of symptoms, the athletes were able to bike several hours before getting attention by the medical crew. They further reported symptom relief during uphill and aggravation of symptoms during downhill biking. To our knowledge, this undulating symptom behavior has yet to be described in the literature and may provide insight into contributing mechanisms to SIPE.

Although SIPE is commonly referred to as a rare condition, there are frequent reports of incidences during triathlon races. We have identified at least 16 reported cases published in the period 2010 to 2019 [6,7,8,9,10]. Symptoms appear to debut with sudden onset of shortness of breath during swimming. This may be severe enough to prompt immediate withdrawal from the race, or may gradually worsen, including hemoptysis, throughout the bike and run phase of a race. When examining 31 asymptomatic Ironmen participants with chest sonography pre- and post-race, Pingitore and colleagues observed subclinical signs of increased pulmonary water content in 23 athletes (74%) [11]. They also found a significant correlation between increased water content and cardiac-related variables and NH_2_-terminal pro-brain natriuretic peptide (NT-proBNP). A study on autopsies following triathlon deaths found a greater proportion of left ventricular hypertrophy (LVH) among deceased triathletes compared to what was expected in the triathlon population [12]. With LVH being a proposed risk factor for SIPE, the authors hypothesize that SIPE may well be a significant contributor to swimming-related deaths in triathlon.

Further contributors to the development of SIPE have been described to include whole-body immersion including face immersion, cold water, use of wet suit and/or swim cap, sudden physical exertion, and any situation that could raise central blood pressure on the race morning, such as excessive hydration or anxiety. Previous studies have hypothesized that SIPE-prone individuals develop hypertension and elevated left ventricular (LV) end-diastolic pressure when exposed to cold, increased oxygen intake and exercise, although more recent studies show that cold exposure is not a prerequisite for developing SIPE [3,13]. Furthermore, elevated levels of cardiac troponin T and abnormal left ventricular function following Ironman races have been suggested as a possible link to pulmonary edema [14].

A unique factor related to immersion in both swimming and diving is the well-documented fluid shift that occurs due to pressure effects on venous blood pooling. Studies have demonstrated a 600 to 700 mL shift of blood from the venous system into the central circulation when immersed to the neck, which in turn increases lung vascular volume and likely contributes to the development of pulmonary edema [15]. This fluid shift may lead to right and LV stroke volume mismatch potentially similar to that seen in other forms of acute heart failure [16]. The Frank-Starling mechanism that normally will counteract the stroke volume imbalance is likely to have reached its physiological limit during a triathlon. This results in the accumulation of fluid in the lungs. Systemic venous constriction and redistribution of blood from the peripheral circulation will maintain the right ventricular filling pressures, and the ongoing stroke volume difference will result in pulmonary edema.

Triathletes typically aim to minimize hydrodynamic drag and to this end wear tightly fitted wetsuits. The potential role of wetsuit use in the development of SIPE is not clearly understood. However, it has been suggested that tight-fitting wetsuits may increase cardiac preload, and thereby, contribute to the development of SIPE [1]. The authors are aware of several SIPE cases where athletes have avoided reoccurrence when replacing tight wetsuits with more loosely fitted suits. Although more research is needed, it may be that the fitting and use of wetsuits is a modifiable risk factor for SIPE.

Once the alveolar–capillary membrane has been disrupted, any elevation in capillary pressure would be expected to facilitate a fluid shift to the alveoli. During exercise, pulmonary vascular pressure remains high [17]. If pulmonary edema is already established, cycling and running may, therefore, maintain or worsen an existing pulmonary edema. Interestingly, all three athletes presented in this report reported symptom relief during uphill cycling and aggravation during downhills. The Norseman Xtreme Triathlon cycling course is characterized by hills of long durations, altitudes of up to 1245 m above sea level and shifting weather conditions. Rain and air temperatures as low as 6 to 10 °C are not uncommon in the more exposed parts of the course. Preliminary results from yet unpublished temperature recordings of the Norseman Xtreme Triathlon participants suggests considerable terrain-associated changes in core temperature during the cycling leg. One could hypothesize that the combination of cold air, high speeds and resulting core temperature drop during downhills result in peripheral vasoconstriction that facilitates increased pulmonary artery pressure. whereas the higher heart rates and increased thermogenic heat produced during uphills might lead to a decrease in peripheral vascular resistance. Along with lower speeds and a more upright riding position, this may reverse a fluid shift to the pulmonary circulation and lung tissue and potentially explain the reported symptom improvement during uphill cycling.

Triathletes are often adept at experiencing discomfort and continuing racing despite fatigue and dyspnea, and a pulmonary edema in development may, therefore, go unrecognized. However, racing in the absence of SIPE usually involves higher exertion in uphills than downhills. The presence of symptom improvement going uphill and worsening dyspnea and cough during downhill cycling may help athletes distinguish SIPE in development from the respiratory stress of physiological exertion.

The presented results underscore that race medics should be familiar with the potential for slow progression and late presentation of SIPE. Biking and running may sustain and worsen an ongoing SIPE, and symptoms, such as facial swelling and coughing, may present very late in the race, especially if the race is hilly. Symptoms might be relieved when going uphill and worsen during downhill sections of the bike ride, and cold weather may aggravate the condition. A focus on symptoms and devices to measure SpO_2_ in the field may facilitate early detection of symptoms of this potentially fatal complication. Symptoms of SIPE usually resolve after normalization of the physiologic environment and by supportive treatment, such as oxygen therapy and occasionally β2-agonists [18]. Furthermore, the athlete should be aware of the increased possibility of recurrent episodes [19].

Summary of take-home messages:Athletes and race crews should recognize the common symptoms of SIPE;These include shortness of breath, cough and blood-stained sputum;Triathletes with SIPE may present for medical examination very late in the race;Symptoms may improve during uphill and worsen during downhill cycling;Pulse oximeters may assist race medics in identifying potential cases of SIPE;SIPE is usually self-limiting upon cessation of exertion, but hospitalization, oxygen therapy or beta2/diuretic therapy may be warranted.

## Figures and Tables

**Figure 1 sports-07-00137-f001:**
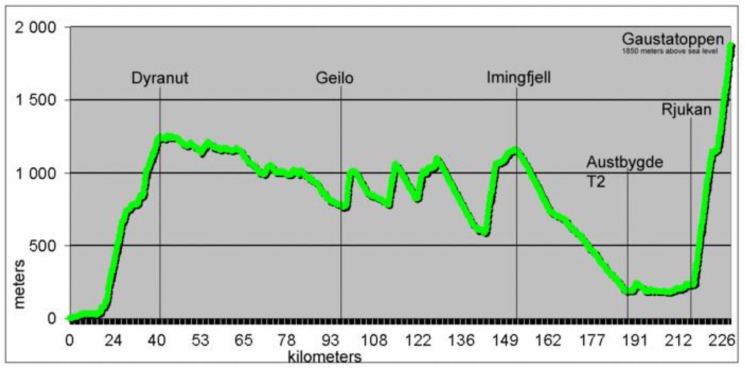
The elevation profile for the Norseman Extreme Triathlon.

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
