# Peer review of "Late-Presenting Swimming-Induced Pulmonary Edema: A Case Report Series from the Norseman Xtreme Triathlon"

_sports, 2019, doi:10.3390/sports7060137_

Round 1
Reviewer 1 Report
Authors reported 3 cases of suspected SIPE during triathlon. The careful clinical findings about the relief and progression of symptoms during uphill and downhill biking are unique. However, several concerns should be warranted.
Major
(1) To the best of my knowledge, the definition of diagnose of SIPE is not established. Compared with scuba divers, triathletes have a chance to be exposed not only immersion but to strenuous exercise through the bike and run. Even though the SIPE was recognized as a dyspnea, it may start gradually (not like cardiogenic pulmonary edema). Therefore, there is a gap between the onset and the abstention. I agree that the reported cases might have been SIPE, but exercise-induced pulmonary edema may be rational diagnosis for them. At least, authors cannot say definitely concrete diagnosis of SIPE because of the late onset of symptoms after immersion.
Authors should carefully use the exercise-induced pulmonary edema and SIPE, and please standardize the terminology of them in the title and discussion.
Minor
(1) A 40-year old female was diagnosed as EIPE or SIPE. But the authors did not describe that the cause of the hypoxia was non-cardiogenic pulmonary edema. Please provide the information how to diagnose pulmonary edema for the case (e.g. X-ray findings). Otherwise, we cannot deny the possibility of differential diagnosis such as angio-edema or asthma attack.
Author Response
Comments and Suggestions for Authors
Authors reported 3 cases of suspected SIPE during triathlon. The careful clinical findings about the relief and progression of symptoms during uphill and downhill biking are unique. However, several concerns should be warranted.
Answer: We appreciate the thorough and constructive feedback on our manuscript. Please see detailed responses below.
Major
(1) To the best of my knowledge, the definition of diagnose of SIPE is not established. Compared with scuba divers, triathletes have a chance to be exposed not only immersion but to strenuous exercise through the bike and run. Even though the SIPE was recognized as a dyspnea, it may start gradually (not like cardiogenic pulmonary edema). Therefore, there is a gap between the onset and the abstention. I agree that the reported cases might have been SIPE, but exercise-induced pulmonary edema may be rational diagnosis for them. At least, authors cannot say definitely concrete diagnosis of SIPE because of the late onset of symptoms after immersion.
Authors should carefully use the exercise-induced pulmonary edema and SIPE, and please standardize the terminology of them in the title and discussion.
Answer: We agree that the lack of absolute diagnostic criteria for the SIPE diagnosis presents a challenge when reporting on suspected SIPE cases. Furthermore, the late presentation to race medics warrants the question of whether symptoms were in fact induced by swimming per se, or simply by the strenuous activity regardless of water immersion. We suggest that in the three presented cases, there can be little doubt that the debut of symptoms occurred during swimming. Furthermore, all three athletes reported symptoms, and changes of said symptoms during early stages, as well as throughout the cycling leg. As pointed out by reviwer 2, once the alveolar-capillary membrane has been disrupted, continued land based exercise might worsen the situation by maintaining an elevated pulmonary vascular pressure and thereby facilitating a shift of fluids through the already disrupted membrane. Due to the relatively early symptom debut, and continuously undulating symptom behavior, we deem it probable that the clinical presentations presented is representative of one continuous series of pathophysiological mechanisms, as opposed to separate events. As such, we would also suggest that “swimming induced pulmonary edema” is the most precise term to describe the presented cases. The use of SIPE/EIPE has been amended accordingly in the revised version of the manuscript.
Minor
A 40-year old female was diagnosed as EIPE or SIPE. But the authors did not describe that the cause of the hypoxia was non-cardiogenic pulmonary edema. Please provide the information how to diagnose pulmonary edema for the case (e.g. X-ray findings). Otherwise, we cannot deny the possibility of differential diagnosis such as angio-edema or asthma attack.
Answer: We agree with the above. As far as we can tell this patient did not receive additional diagnostic investigations. On the basis of the clinical examination of the on-site race doctor we still deem SIPE the most likely diagnosis. We have amended the revised version of the manuscript with a rationale for this argument. Additionally, the text has been altered to better reflect the level of uncertainty pertaining to the suspected diagnosis in this case.
Reviewer 2 Report
This is a well-described series of 3 cases characteristic of SIPE occurring during a triathlon, where each person proceeded with the bike after onset of symptoms during the swim, resulting in. This series illustrates further that if the victim persists in biking/running after onset of pulmonary edema manifestations during the swim phase, SIPE can worsen. The important message for race organizers and physicians is that triathletes may present for evaluation and treatment of SIPE even during the 'dry' phases of a triathlon.
A few minor points:
Title: These cases are not truly delayed onset: lines 67-69 state: "They later reported that they felt increasingly breathless and was struggling to keep the usual pace in the water just minutes after the start." I suggest amending the title to reflect 'persistence', 'worsening', etc.
Lines 52-3: "All three cases were diagnosed with swimming-induced pulmonary edema (SIPE) and displayed behaviour not previously described in the literature". Triathletes are tough to the point of 'working through' adverse symptoms, and persistence with the biking phase after symptom onset has been described at least once (Carter EA, Koehle MS. Immersion pulmonary edema in female triathletes. Pulm Med. 2011;2011:261404. doi:10.1155/2011/261404. Epub 2011 Jun 1).
Please include the normal range for for BNP, troponin, CRP and D-dimer.
Arterial blood gas values are either reported either erroneously or with the incorrect units. Are these values kPa rather than mmHg?
I suggest emphasizing that once pulmonary edema occurs it could be worsened by biking/running, because during exercise pulmonary vascular pressures will remain high (Naeije R, Chesler N. Pulmonary circulation at exercise. Compr Physiol. 2012 Jan;2(1):711-41). Once the alveolar-capillary membrane has been disrupted, any elevation in pulmonary capillary pressure will facilitate alveolar fluid/blood accumulation.
Author Response
Comments and Suggestions for Authors
This is a well-described series of 3 cases characteristic of SIPE occurring during a triathlon, where each person proceeded with the bike after onset of symptoms during the swim, resulting in. This series illustrates further that if the victim persists in biking/running after onset of pulmonary edema manifestations during the swim phase, SIPE can worsen. The important message for race organizers and physicians is that triathletes may present for evaluation and treatment of SIPE even during the 'dry' phases of a triathlon.
Answer: We appreciate the thorough feedback on our manuscript and the engagement reflected in your review. Please see detailed responses below.
A few minor points:
Title: These cases are not truly delayed onset: lines 67-69 state: "They later reported that they felt increasingly breathless and was struggling to keep the usual pace in the water just minutes after the start." I suggest amending the title to reflect 'persistence', 'worsening', etc.
Answer: We agree. The title and revised version of the manuscript has been amended accordingly.
Lines 52-3: "All three cases were diagnosed with swimming-induced pulmonary edema (SIPE) and displayed behaviour not previously described in the literature". Triathletes are tough to the point of 'working through' adverse symptoms, and persistence with the biking phase after symptom onset has been described at least once (Carter EA, Koehle MS. Immersion pulmonary edema in female triathletes. Pulm Med. 2011;2011:261404. doi:10.1155/2011/261404. Epub 2011 Jun 1).
Answer: This phrase was intended to refer to the undulating nature of the presenting symptoms during the bike course. We agree this was not made clear in the above wording. The revised version of the manuscript has been amended to specify this point.
Please include the normal range for for BNP, troponin, CRP and D-dimer.
Answer: Normal range values for the above tests have been added to the revised version of the text.
Arterial blood gas values are either reported either erroneously or with the incorrect units. Are these values kPa rather than mmHg?
Answer: We agree. These values were given with incorrect units. The correct units are, as you suggest, kPa. This has now been corrected in the revised version of the manuscript.
I suggest emphasizing that once pulmonary edema occurs it could be worsened by biking/running, because during exercise pulmonary vascular pressures will remain high (Naeije R, Chesler N. Pulmonary circulation at exercise. Compr Physiol. 2012 Jan;2(1):711-41). Once the alveolar-capillary membrane has been disrupted, any elevation in pulmonary capillary pressure will facilitate alveolar fluid/blood accumulation.
Answer: We agree this is a great point. Additions have been made to the discussions section of our revised manuscript to address this.
Reviewer 3 Report
This is a very important topic to bring awareness to.
Overall comments. There are some minor grammatical issues that need to be addressed to help with the reading of the paper.
Presentation of units needs to be corrected. A space provided between the number and all units except for a % sign.
During the case report section. I query the reporting of all the clinical tests and results as these results are not discussed further. I am however, not a medical doctor and these tests may be significant to them. To extend the range of readership I would suggest putting a brief note regarding the results - for example results ruled out other diagnoses.
I would suggest that the paper ends with a summary box of take home message. Such as symptoms to be aware of both for triathletes (so they don't keep trying to finish the race), and symptoms and responses for race medics. I believe this is an important issue and a take home message box would provide a concise summary of recommendations that limits the readers work to find the message.
Having experienced SIPE myself - I noted that the incident occurred following pre-hydration and coincided with wearing a 'properly fitted' wetsuit. I now wear a looser wetsuit and have had no further incidence even with prehydration. I do not know if there is other evidence of this, but perhaps a discussion around why wearing a wetsuit might increase the risk would help those who have experienced SIPE modify risk factors in order to continue to compete.
Author Response
Comments and Suggestions for Authors
This is a very important topic to bring awareness to.
Answer: We appreciate the thorough and constructive feedback on our manuscript. Please see detailed responses below.
Overall comments. There are some minor grammatical issues that need to be addressed to help with the reading of the paper.
Answer: We agree this would help with reading of the paper. The revised version of the manuscript has been proof-read and numerous grammatical errors have been amended.
Presentation of units needs to be corrected. A space provided between the number and all units except for a % sign.
Answer: Agreed. This has now been corrected in the revised version of our text.
During the case report section. I query the reporting of all the clinical tests and results as these results are not discussed further. I am however, not a medical doctor and these tests may be significant to them. To extend the range of readership I would suggest putting a brief note regarding the results - for example results ruled out other diagnoses.
Answer: We agree this would be beneficial information for the reader. The discussion part of our revised manuscript has been amended to include comments on test results and diagnostic considerations.
I would suggest that the paper ends with a summary box of take home message. Such as symptoms to be aware of both for triathletes (so they don't keep trying to finish the race), and symptoms and responses for race medics. I believe this is an important issue and a take home message box would provide a concise summary of recommendations that limits the readers work to find the message.
Answer: A summary box has been added to the revised version of the manuscript.
Having experienced SIPE myself - I noted that the incident occurred following pre-hydration and coincided with wearing a 'properly fitted' wetsuit. I now wear a looser wetsuit and have had no further incidence even with prehydration. I do not know if there is other evidence of this, but perhaps a discussion around why wearing a wetsuit might increase the risk would help those who have experienced SIPE modify risk factors in order to continue to compete.
Answer: The revised version of the manuscript has been altered to include a brief discussion on the potential contribution of wetsuits to the development of SIPE.
Round 2
Reviewer 1 Report
All comments have been considered. This revised article becomes much better than the latest one. Thank you for your revision.
Author Response
Thank you very much for your valuable comments to our manuscript. We indeed agree that the manuscript is much better after the first round of review. You helped us make this case report better!